



# 1 Dynamic changes of terrestrial net primary production and its

# 2 feedback to evapotranspiration

Zhi Li[1], Yaning Chen[1], Yang Wang[2] and Gonghuan Fang[1,3]
[1] State Key Laboratory of Desert and Oasis Ecology, Xinjiang Institute of Ecology and Geography, Chinese Academy of
Sciences, Urumqi 830011, China
[2] College of Pratacultural and Environmental Sciences, Xinjiang Agricultural University, Urumqi 830052, China
[3] Department of Geography, Ghent University, Ghent, Belgium
*Correspondence to*: Yaning Chen (chenyn@ms.xjb.ac.cn)
**Abstract**. Earth experienced dramatic environmental changes in the recent 15 years (2000-2014). The
past decade has been the warmest in the instrumental record, which significantly influences the global
water cycle and vegetation activities. Overall, the global inter-annual series of net primary production
(NPP) slightly increased in 2000-2014 at a rate of 0.06 PgC/yr$^2$. More than 64% of vegetated land in the
Northern Hemisphere showed increased net primary production, while 60.3% of vegetated land in the
Southern Hemisphere showed decreased trend. Net primary production correlates positively with land
actual evapotranspiration (ET), especially in the Northern Hemisphere, where the increased vegetation
productivity (0.13 PgC/yr$^2$) promotes decadal rises of terrestrial evapotranspiration (0.61 mm/yr$^2$).
However, anomalous dry conditions led to reduced vegetation productivity (-0.18 PgC/yr$^2$) and nearly
ceased growth in terrestrial evapotranspiration in the Southern Hemisphere (0.41 mm/yr$^2$). Under the
content of past warmest 15 years, global potential evapotranspiration (PET) shows an increasing trend





of 1.72 mm/yr$^2$, while precipitation for the domain shows a variability positive trend of 0.84 mm/yr$^2$,
which consistent with expected water cycle intensification. But precipitation trend is lower than
evaporative demand, indicating some moisture deficit between available water demand and supply for
evapotranspiration, thereby accelerated soil moisture loss. Drought indices and
precipitation-minus-evaporation suggested an increased risk of drought in the present century.
To understand why climates in the northern and southern hemispheres respond differently to NPP, the
results showed that temperature is the dominant control on vegetation growth in the high latitude in the
Northern Hemisphere, while net radiation is the main effect factors to NPP in the mid latitude, and in
arid and semi-arid biomes also mainly driven by precipitation. While in the Southern Hemisphere, NPP
decreased because of warming associated drying trends of PDSI.



## 1 Introduction

Organizations such as the Intergovernmental Panel on Climate Change (IPCC) and the World Meteorological Organization (WMO) have reported that the recent decade was the warmest on record. Warming indicating a general prospective acceleration or intensification of the global hydrological cycle and thus alters evapotranspiration (Wentz et al., 2007; Douville et al., 2013), with implications for the response and mutual feedbacks of ecosystem services (Jung et al., 2010; Davie et al., 2013). Most of the research analyzed the impacts of climate change on vegetation activities, and some research on feedback of terrestrial ecosystems to climate change has focused on their potential role as carbon sources (Field et al., 2007). Fewer study revealed the feedback of vegetation interannual variability on the land surface physical processes and climate system (Zhi et al., 2009), especially on evapotranspiration.

Terrestrial net primary production (NPP), defines as the amount of photosynthetically fixed carbon available to the first heterotrophic level in an ecosystem, links terrestrial biota with the atmosphere system (Beer et al., 2010). There have been considerable efforts of ecosystem models to estimate terrestrial NPP, owing to its importance for ecological and social systems at continental and global scales (Chen et al., 2012; Potter et al., 2012; Pan et al., 2014). Several studies have shown that climate constraints (e.g. with increasing temperature and solar radiation) were relaxing (Nemani et al., 2003). The interaction of temperature, radiation and water has imposed complex and varying limitations on vegetation activities in different regions of the world. The spatial variation of NPP depends on regional soil and climatic conditions, vegetation types, and human activities, while the temporal variation of NPP



depends on the annual and seasonal variability of climatic factors. The temporal-spatial variation and
attribution in global terrestrial NPP under the content of high variability warming are still lacking.

In turn, vegetation productivity influences albedo and emissivity, which strongly regulates global

climate (Chapin et al., 2011), which is especially obvious in land evapotranspiration. Land
evapotranspiration is a central process in the climate system and a nexus of the water, energy and
carbon cycles. Hence, it plays a pivotal role in maintaining the water and heat balance. Shen and
colleagues (2015) reported that, in contrast with the Arctic region (i.e., positive feedback to warming),
increased vegetation activity may attenuate daytime warming by enhancing actual evapotranspiration as
a cooling process on the Tibetan Plateau. Zhang et al. (2015) investigated that climate change and recent
vegetation greening promote multi-decadal rises of global land evapotranspiration, while anomalous
drought between 2000 and 2009 led to reduced vegetation productivity in the Southern Hemisphere
(Zhao and Running, 2010). However, little observational evidence exists to demonstrate vegetation
feedback on climate in different global geographical units.

Investigating factors that control changes in NPP and its feedback effects could provide important

clues to the underlying mechanisms and the complex interactions between ecosystems and climate
systems (Tian et al., 2000; 2012). Having a clear understanding of the land's biophysical feedback to the
atmosphere is crucial if we are to simulate regional climate accurately. In our study, we investigated: 1)
whether the high volatility temperature of the past decade continued to increase NPP, or if different
climate constraints were at play. 2) why climates in the northern and southern hemispheres respond
differently to NPP? 3) what is the temporal-spatial variation of NPP and its feedback to
evapotranspiration?



## 2 Data and Methodology

### 2.1 Data

The monthly grid data of the temperature and precipitation series (2000-2014), with the spatial resolution of 0.5 degree, were collected from the Climatic Research Unit (http://www.cru.uea.ac.uk/data/).

The radiation and soil moisture data series were come from Global Land Data Assimilation System (GLDAS-1), with the spatial resolution of 0.25 degree (http://gdata1.sci.gsfc.nasa.gov/daac-bin/ G3/gui.cgi?instance_id=GLDAS025_M). The depths of the four soil layers are: 0-10 cm, 10-40 cm, 40-100 cm, and 100-200 cm. The quality of the GLDAS data set was assessed against available observations from multiple sources (Zhang et al., 2008; Chen et al., 2013).

The monthly data of Palmer Drought Severity Index (PDSI), with the spatial resolution of 2.5 degree, was available at http://www.cgd.ucar.edu/cas/catalog/ climind/pdsi.html. PDSI, as a indicator of land surface moisture conditions, has been widely used in routinely monitoring and assessing global and regional drought conditions. The global dry areas were defined as PDSI < -3.0, while the wet areas were defined as PDSI > + 3.0 (Dai et al., 2004).

We used the Global Land Cover Characterization data from the International Geosphere-Biosphere Program (IGBP) in 2000 (http://edc2.usgs.gov/glcc/glcc.php), and MODIS in 2000 and 2013 (http://modis.gsfc.nasa.gov/data/dataprod/mod12.php). From these data, a routinely integrated classification of land use/cover change (LUCC) characteristics can be obtained based on the feature fusion processes.



We unified the spatio-temporal resolution of these data from different sources based on the
re-sampling and re-classification techniques.
**2.2 Methods**
*NPP algorithm.* Net primary production estimations are typically model-based and biogeochemical,
generated from a larger set of simulated C fluxes between the atmosphere and terrestrial ecosystems (Ito
et al., 2011). The global 1-km MODIS NPP datasets from 2000 to 2014 are from MOD17. A better
agreement of MODIS and terrestrial NPP estimates allows for the use of MODIS in large-scale
estimates (Neumann et al., 2015). The algorithm calculates annual NPP as:
$$NPP = \sum_{i=1}^{365} (GPP - R_m) - R_g \qquad (1)$$
Similarly, the algorithm calculates daily GPP as:
$$GPP = \varepsilon_{max} \times SW_{rad} \times FPAR \times fVPD \times fT_{min} \qquad (2)$$
$R_m$ is the maintenance respiration, which is a function of daily average temperature ($T_{avg}$):
$$R_m = Q_{10}^{(\frac{T_{avg}-20}{10})} \qquad (3)$$
$$Q_{10} = 3.22 - 0.046 \times T_{avg} \qquad (4)$$
Therefore,
$$NPP = \sum_{i=1}^{365} (GPP - R_m) - R_g = \sum_{i=1}^{365} (GPP - R_m) - 0.25 \times NPP \qquad (5)$$
which means:


$$NPP = 0.8 \times \sum_{i=1}^{365} (GPP - R_m), \quad \text{where} \quad \sum_{i=1}^{365} (GPP - Rm) \geq 0 \qquad (6)$$
$$NPP = 0, \quad \text{where} \quad \sum_{i=1}^{365} (GPP - Rm) < 0 \qquad (7)$$
where $\varepsilon_{max}$ is the maximum light use efficiency, $SW_{rad}$ is short-wave downward solar radiation (of
which 45% is Photosynthetically Active Radiation (PAR)), FPAR is the fraction of PAR being absorbed
by the plants, fVPD and $fT_{min}$ are the reduction scalar from high daily time Vapor Pressure Deficit and
low daily minimum temperature ($T_{min}$), respectively, and annual growth respiration ($R_g$) is a function of
annual maximum leaf area index (LAI). Zhao and Running (2010) modified the calculations by
assuming that growth respiration is approximately 25% of NPP.
*ET and PET algorithm.* The MODIS evapotranspiration datasets are estimated using Mu and
colleagues (2011) improved ET algorithm over Mu et al.'s (2007) previous paper. Based on the
energy-balance theory and the Penman-Monteith equation, the required MODIS data inputs ET
algorithms, including daily meteorology (temperature, actual vapor pressure, and incoming solar
radiation) remotely-sensed land cover, FPAR/LAI, and albedo (Friedl et al., 2002, 2010; Myneni et al.,
2002; Jin et al., 2003). The output variables include evapotranspiration (ET), latent heat flux (LE),
potential ET (PET), potential LE (PLE) and quality control (ET_QC).
*Trend analysis.* To further discern the trends of yearly NPP and ET, we examined linear trends
estimation on a per-pixel basis to establish a linear regression relationship between variables ($x_i$) and
time ($t_i$). The regression coefficient (b) is:



$$b = \frac{n \times \sum\limits_{i=1}^{n} x_i t_i - \sum\limits_{i=1}^{n} x_i \sum\limits_{i=1}^{n} t_i}{n \times \sum\limits_{i=1}^{n} t_i^2 - \left(\sum\limits_{i=1}^{n} t_i\right)^2} \qquad (8)$$

*Partial correlation analysis.* This method is used to describe the relationship between two variables
while taking away the effects of several other variables. The partial correlation of $x_1$ and $x_2$ is adjusted
for a third variable y (at a significance level of 0.05 by t-text):
$$r_{x1x2 \cdot y} = \frac{r_{x1x2} - r_{x1y} r_{x2y}}{\sqrt{\left(1 - r_{x1y}^2\right)\left(1 - r_{x2y}^2\right)}} \qquad (9)$$

**3 Results and analysis**
**3.1 Temporal-spatial variation in global terrestrial NPP and its feedback to ET**
The spatial pattern of global NPP trend (2000-2014) steadily decreasing from the equator to the Arctic
and Antarctic (Fig. 1c). Overall, the inter-annual series of NPP increased moderately at a rate of 0.06
PgC/yr$^2$ throughout the last 15 years. Additionally, it shows different changes in the Northern
Hemisphere (NH) and Southern Hemisphere (SH). While increasing over large areas in the NH (Fig. 1a),
decreased in the SH (Fig. 1b). Specifically, in the NH, 64% of vegetated area had increased NPP,
including large areas of North America, Western Europe, India, and the eastern China. Regions with
decreased NPP include east Europe, high latitudes of central and west Asia. In the SH, decreased NPP
accounted for about 60.3% of vegetated land area, mainly concentrated in the large parts of South
America, south Africa, and west Australia. Furthermore, in the equatorial regions, Amazon rainforests



had significantly decreased NPP, whereas African rainforests experienced an increasing trend (Fig. 1c).
Because tropical rainforest NPP accounts for a large proportion of global NPP, decreases in the SH
partially counteracted increases in NH.
By combining the global land use/cover change (LUCC) characteristics, the results showed that
shrubland has the greatest potential increasing trend of NPP (16.5 gC/m$^2$/yr) compared to other biomes,
followed by grassland (12.5 gC/m$^2$/yr). This may be related to the expansion of woody vegetation over
the past 15 years. In Arctic tundra (Hughes et al., 2006) and lower latitudes in arid environments (Chen
et al., 2014), experimental studies provided clear evidence that climate warming is sufficient to account
for the expansion of shrubs and graminoids.
Changes in vegetation albedo and emissivity exert feedback on climate, which is especially obvious
in evapotranspiration (Field et al., 2007). Increased vegetation productivity and climate change promote
multi-decadal rises of global land ET (Zhang et al., 2015). The average mean of estimated global annual
ET is 518.6 mm/yr, with an inter-annual trend of 0.46 mm/yr$^2$. Figure 1a&1b showed that NPP
correlates positively with ET (especially in the NH, however, reduced vegetation productivity nearly
ceased growth in ET in SH). The variations anomaly in the SH has a much higher variability, so the
consistency between NPP and ET in the SH is less than that in the NH. The spatial inconsistency in the
SH mainly occurred near the equator, e.g., southern African rainforests (Fig. 1b&1d). These regions
have high values of average annual precipitation and stronger variability than elsewhere, causing greater
changes to ET and its components (land surface evaporation, canopy evaporation and transpiration).
When the inter-annual variability of NPP is small, the ET component of land surface evaporation
increases. In contrast, in areas with large inter-annual variability of NPP such as shrubland and grass



dominant regions, the ET components of land surface evaporation declines and transpiration increases.

## 3.2 Major climatic control factors to NPP variation

To understand why climates in the northern and southern hemispheres respond differently to NPP, we
first estimated the spatial trends of climatic control factors, and then analyzed the complex multiple
climatic constraints to plant growth. A comprehensive interpretation of interactive climatic controls on
plant productivity showed that water, temperature and radiation are the key factors affecting vegetation
growth. Globally, growth was most strongly limited by water availability on 40% of Earth's vegetated
surface, while temperature limitations exerted the main controlling influence on 33%, and radiation on
27% (Nemani et al., 2003).
From 2000 to 2014, overall trends of average annual temperature (0.007 ℃/yr$^2$) and precipitation
(0.84 mm/yr$^2$) experienced world-wide increases while showing different temporal change patterns (Fig.
2a & 2b). Eastern Europe, South America, southern Africa and western Australia experienced warming
combined with decreased precipitation (warm-dry trend), whereas southeast North America, western
Europe, east Russia, and African rainforests experienced warming combined with increased
precipitation (warm-wet trend). Meanwhile, net radiation (Rn) increased in the equatorial tropics and
arid regions in Northwestern China, but decreased in the Arctic and Antarctic (Fig. 2c). The Palmer
Drought Severity Index (PDSI) is a widely-used drought index that correlates with soil moisture during
warm seasons (Dai et al., 2004; 2013). Generally, a lower PDSI implies a drier climate. Global PDSI
has decreased at a rate of 0.04/yr$^2$ over the past 15 years. This suggests an increased risk of drought in
the twenty-first century. The spatial trend of PDSI shows that the eastern and northern coasts of North



America, along with the African continent, Eurasia and southern South America, had obvious drought
trends from 2000-2014. Northern China, parts of Mongolia, and western Russia near Lake Baikal also
experienced a drying trend (Fig. 2d). Warming-induced drying resulted from increased ET, and was
most prevalent over NH mid-high latitudes. Drought develops with periods of low accumulated
precipitation and is exacerbated by high temperatures.
We analyzed partial correlations between NPP and temperature (T), precipitation (P), Net radiation
(Rn) and PDSI during growing seasons to determine their respective contributions across different
regions (Fig. 3, Table 1). Climate changes from 2000 to 2014 have made temperature, precipitation and
radiation somewhat beneficial to plant growth. In the NH, climatic changes have eased multiple climatic
constraints to plant growth in an earlier spring, but the continuous warming may offset the benefits.
In NH high latitudes (>47.5 °N), temperature has a positive correlation with NPP (R=0.6). Significant
warming generally lengthens the vegetation growing seasons and promotes plant growth in tundra
regions, so the recent warming in this region has increased NPP (Fig. 3a). For northern mid and low
latitudes (<47.5 °N), where large areas are classified as having an arid climate, vegetation is short-rooted.
NPP has a significant negative correlation with P (R=0.7, p<0.05) (Fig. 3b) and is also correlated to Rn
(Fig. 3c). In areas of high elevation such as the Tibetan plateau (which is similar to high latitudes),
temperature is the dominant control factor in vegetation growth.
Equatorial Amazon rainforests experienced significantly decreased NPP, whereas African rainforests
exhibited an increasing trend. These changes in equatorial regions are mainly related to the warming in
the Amazon along with increasing precipitation in African rainforests. High temperatures cause higher
rates of evapotranspiration, generally reducing soil water availability for vegetation in Amazon.



For the SH, we noted a significant correlation ($r = 0.7$, $p < 0.05$) between NPP and PDSI (Fig. 3d). The
warming trend induced a much higher evaporative demand and led to a drying trend, except for the
afore-mentioned increased precipitation in African rainforests. The PDSI in African rainforests also
showed a slight increasing trend. The general drought event across the SH, which was induced by
extreme heat and precipitation deficit, has resulted in a net water availability reduction. Warming
associated drying directly caused the significantly decreasing trend of NPP in SH.

## 3.3 Continuation feedback of NPP to evapotranspiration likely exacerbate regional drought

Drought indices suggested an increased risk of drought in the present century. We used potential
evapotranspiration (PET) as a surrogate measure of atmospheric moisture demand. Potential
evapotranspiration is defined as the maximum quantity of water capable of being evaporated from the
soil and transpired from the vegetation, and actual evapotranspiration is the actual evaporation from
water and soil, and transpiration from vegetation. Penman (1948) stated that ET had a proportional
relationship with PET, and Bouchet (1963) hypothesized that a complementary feedback mechanism
exists between ET and PET in water-limited regions. Overall, our investigations do indicate that there is
a proportional behavior between ET and PET in humid regions and a complementary one in arid regions
(Fig. 1d and Fig. 4a). PET, combined impacts of temperature, solar radiation, vapor pressure and wind
speed (Zhang et al., 2015), has an interaction process with NPP (Fig. 4b).
Global PET shows an increasing trend of 1.72 mm/yr$^2$ over the past 15-year record, while P for the
domain shows a variability positive trend of 0.84 mm/yr$^2$. It indicates some moisture deficit between
available water demand and supply for evapotranspiration. P is mostly being lost to ET rather than being



allocated to other components of energy and water cycle (Zhang et al., 2015).
Soil moisture is an important sensor for measuring surficial wetness and dryness levels, which almost
reflects the dryness and wetness of climate. With precipitation being the most direct factor influencing
on soil moisture, temperature and solar radiation etc. mainly through evapotranspiration to cause soil
moisture loss. Available soil moisture is defined as the amount of water a plant can access in its root
zone. Thus, spatial and temporal variations in soil moisture closely related to vegetation growth (Davis
and Pelsor, 2001; Yang et al., 2010). Figure 5 illustrates the world-wide decrease in soil moisture in four
layers (0-10, 10-40, 40-100, and 100-200 cm). The increasing soil moisture limitation is a classic
eco-hydrologically-confined factor.

## 4 Discussions

Earth experienced dramatic environmental changes in the recent 15 years of the $21^{st}$ century.
Although a relatively short time series analysis (2000-2014), a strong variation of NPP and its feedback
to evapotranspiration, as well as the correlation with the dramatic climate changes were found
worldwide. There are some uncertainties in the feedbacks of ecosystem responses to evapotranspiration,
but understanding the land-surface ecological feedbacks to the atmosphere processes is necessary if we
are willing to simulate climate change accurately. Several studies showed that the relaxed climate
constraints with increasing temperature and solar radiation, allowed an increased trend in global NPP
over 1982-1999 (Nemani et al., 2003). This was followed by a drought-induced reduction in global NPP
in 2000-2009 (Zhao and Running, 2010). Our study used global 1-km MODIS NPP datasets from 2000
to 2014. The results showed that under the content of past warmest 15 years, the slightly increased



inter-annual series of NPP promote decadal rises of global land ET, thereby accelerated soil moisture

loss. Weather systems can lead to droughts by suppressing precipitation (Beaumont et al., 2011) and by

warming and drying soil via soil-temperature feedback (Seneviratne et al., 2010; Sheffield et al., 2012;

Orlowsky and Seneviratne, 2013; Williams et al., 2014). Drought indices and precipitation-minus

-evaporation suggested an increased risk of drought in the present century.

As noted previously, vegetation feeds back to the spatio-temporal characteristics of climate through

evapotranspiration. Evapotranspiration is a key process that dissipates the energy and water absorbed by

the vegetation and determines the diurnal cycle of near-surface temperature. It is limited mostly by

energy in humid and semi-humid areas, whereas low-value evapotranspiration is limited mostly by

water in arid and semi-arid areas. The different values of average precipitation and variability as well as

different land types will cause diverse changes in evapotranspiration and its components (land surface

evaporation, canopy evaporation and transpiration), thereby producing different feedback to temperature.

There are still major gaps in our understanding of how the responses of terrestrial ecosystems eliminate

or increase the risk of dangerous climate change, and these gaps need to be filled.

**5 Conclusions**

The inter-annual series of global NPP slightly increased for the last 15 years but has different changes

in the Northern Hemisphere and Southern Hemisphere. Over 64% of vegetated land areas had increased

NPP in the Northern Hemisphere while 60.3% had decreased NPP in the Southern Hemisphere. In the

Northern Hemisphere, temperature is the dominant control on vegetation growth in the high latitude,

while net radiation is the main effect factors to NPP in the mid latitude, and in arid and semi-arid



biomes also mainly driven by precipitation. In the Southern Hemisphere, NPP decreased because of
warming associated drying trends of PDSI.
NPP to actual evapotranspiration are likely to be positive feedback, especially significant in the
Northern Hemisphere, where the increased vegetation productivity (0.13 PgC/yr$^2$) reduces albedo,
promotes decadal rises of actual evapotranspiration (0.61 mm/yr$^2$). However, dry conditions led to
reduced vegetation productivity (-0.18 PgC/yr$^2$) and nearly ceased growth in evapotranspiration in the
Southern Hemisphere. Continuation of these trends will likely exacerbate regional drought levels.
**Author Contribution**
Zhi Li and Yaning Chen wrote the main manuscript text, YangWang and Gonghuan Fang prepared
figures 4&6. All authors reviewed the manuscript.
**Acknowledgements**
The research is supported by the CAS "Light of West China" Program (2015XBQNB17) and the
Foundation of State Key Laboratory of Desert and Oasis Ecology (Y471166).

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





**Table:**

Table 1. Correlations between NPP and climatic variables for over both hemispheres

| Zones | NPP trend | T trend | P trend | Rn trend | PDSI trend |
|---|---|---|---|---|---|
| NH high latitudes (>47.5 ˚N) | y=0.02x+30.51 | y=0.021x-5.75  **R= 0.60*** | y=0.104x+46.58  R=0.29 | y=-5.21x+453.6  R=0.45 | y=-0.005x+0.23  R=0.44 |
| NH mid/low latitudes (<47.5 ˚N) | y=0.07x+45.68 | y=0.009x+18.3  R= -0.17 | y=0.341x+76.8  **R= 0.70**** | y=3.239x+105.9  R=0.50 | y=0.006x-0.46  R=0.56 |
| South Hemisphere | y=-0.18x+78.37 | y=0.010x+21.6  R= -0.53 | y=0.074x+116.8  R=0.37 | y=2.455x+129.4  R=0.43 | y=-0.042x+0.33  **R= 0.70**** |





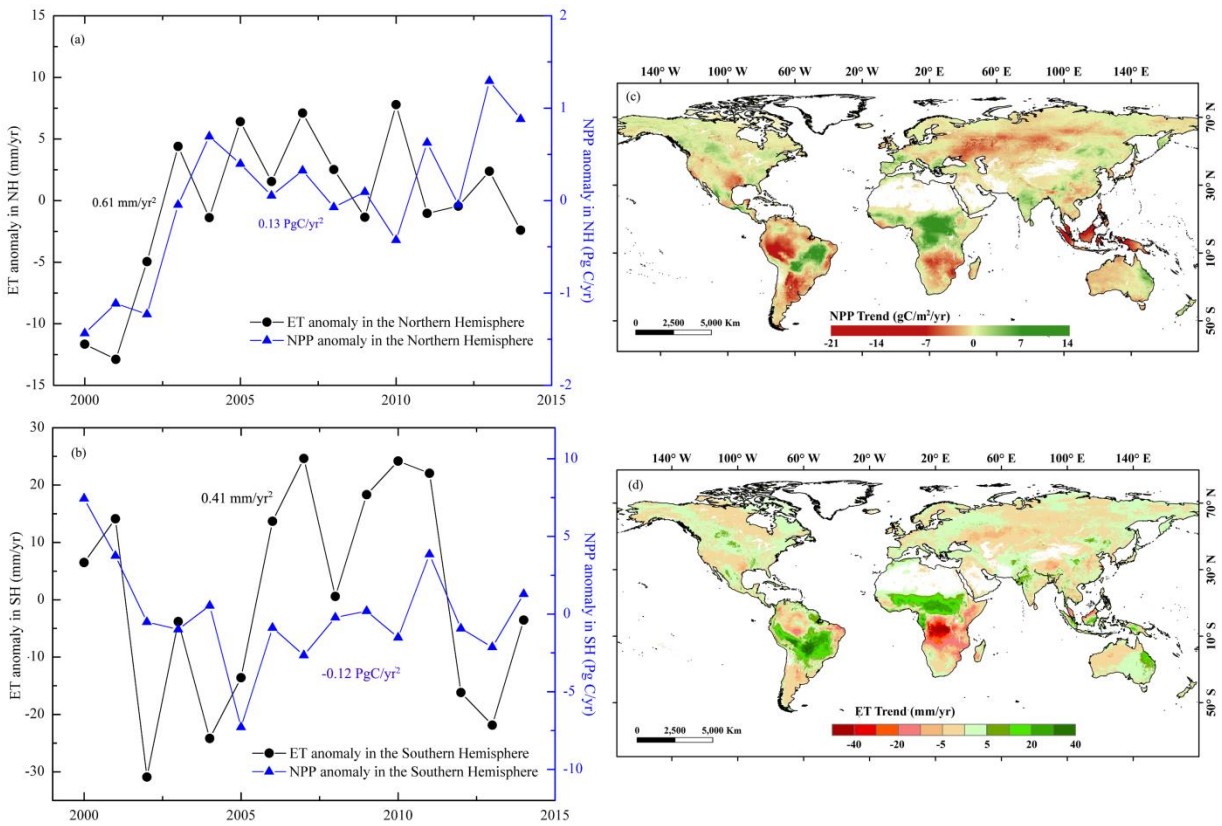


Fig. 1. Temporal-spatial variations in global terrestrial NPP and ET from 2000-2014. (a) Inter-annual

variations of NPP and ET in the Northern Hemisphere (NH). (b) Inter-annual variations of NPP and ET

in the Southern Hemisphere (SH). (c) Spatial pattern of NPP trend. (d) Spatial pattern of ET trend.





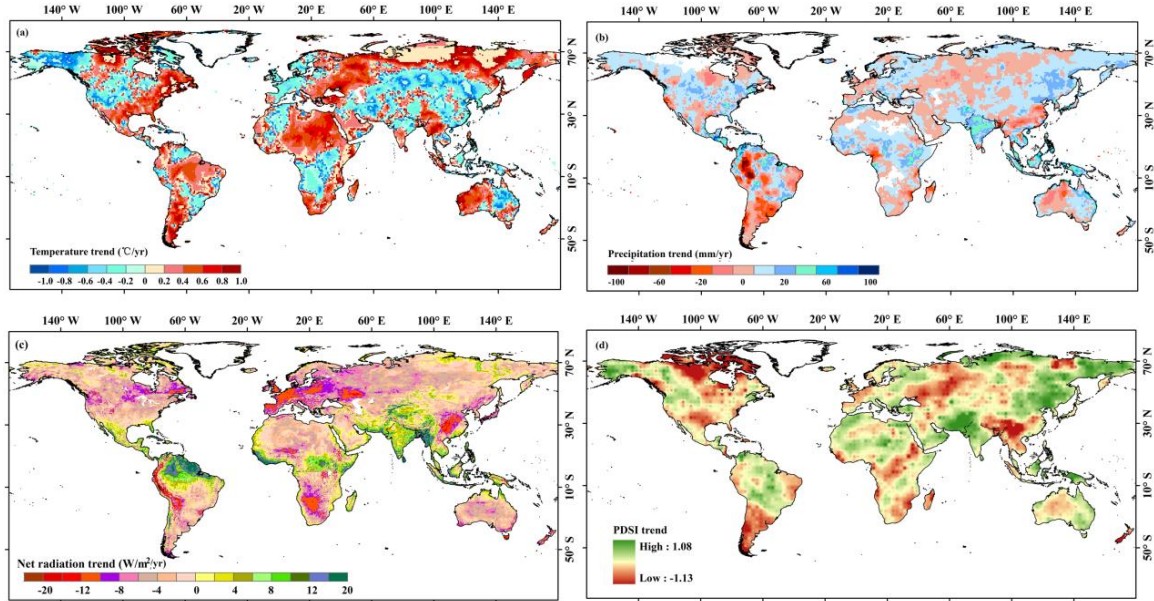


Fig. 2. Trends of air temperature (T), precipitation (P), net radiation (Rn) and Palmer Drought Severity

Index (PDSI) from 2000-2014.





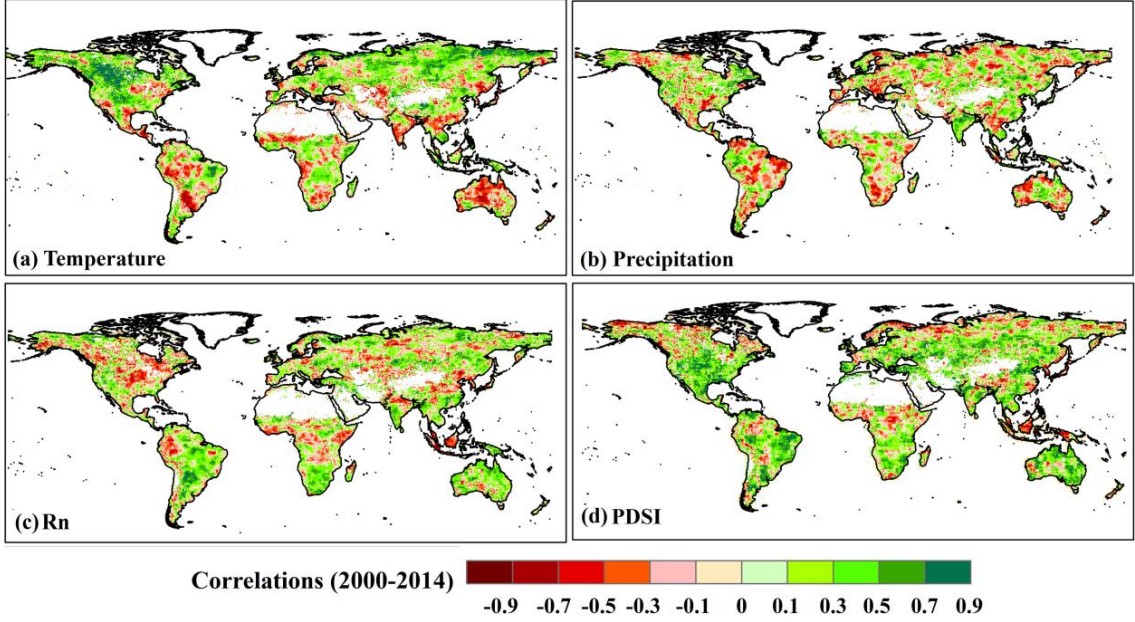


Fig. 3. Partial correlations between NPP and (a) Temperature, (b) Precipitation, (c) Net radiation, (d)

PDSI in growing season.





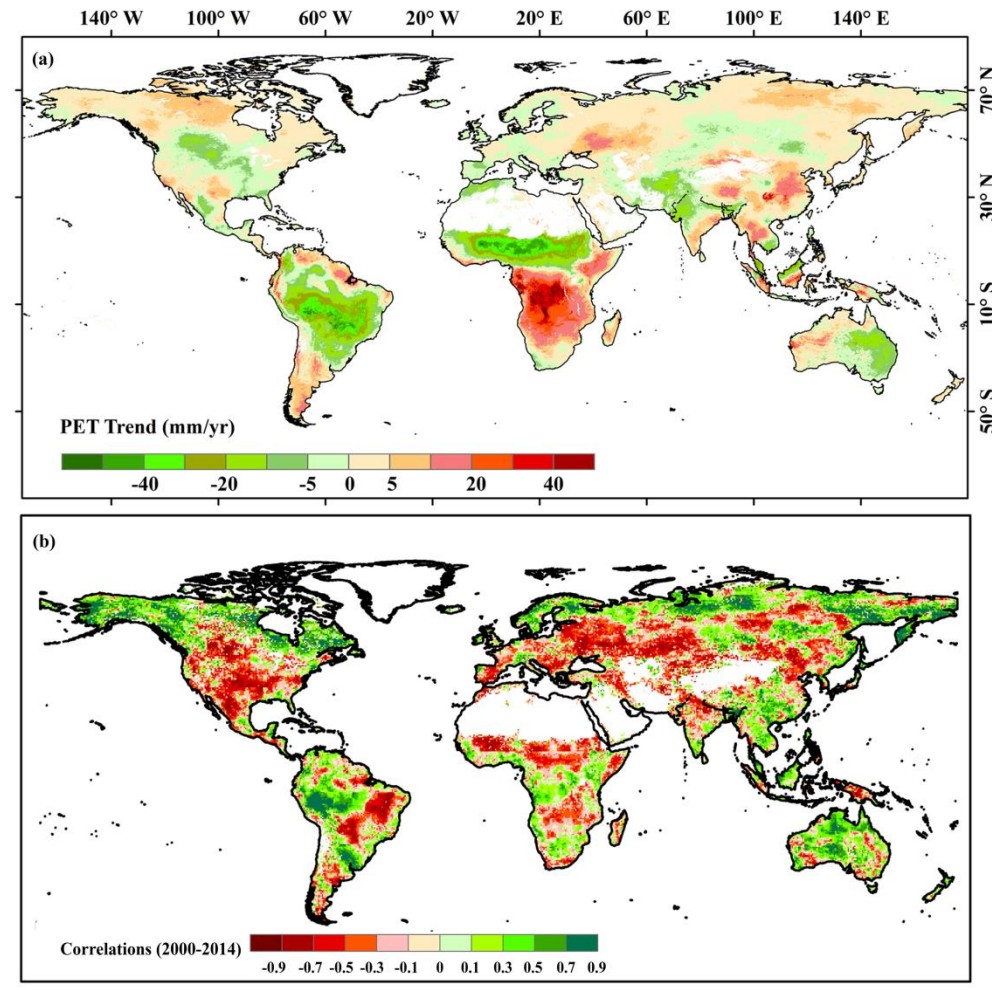


Fig. 4. (a) Spatial pattern of PET trend. (b) Partial correlations between NPP and PET.





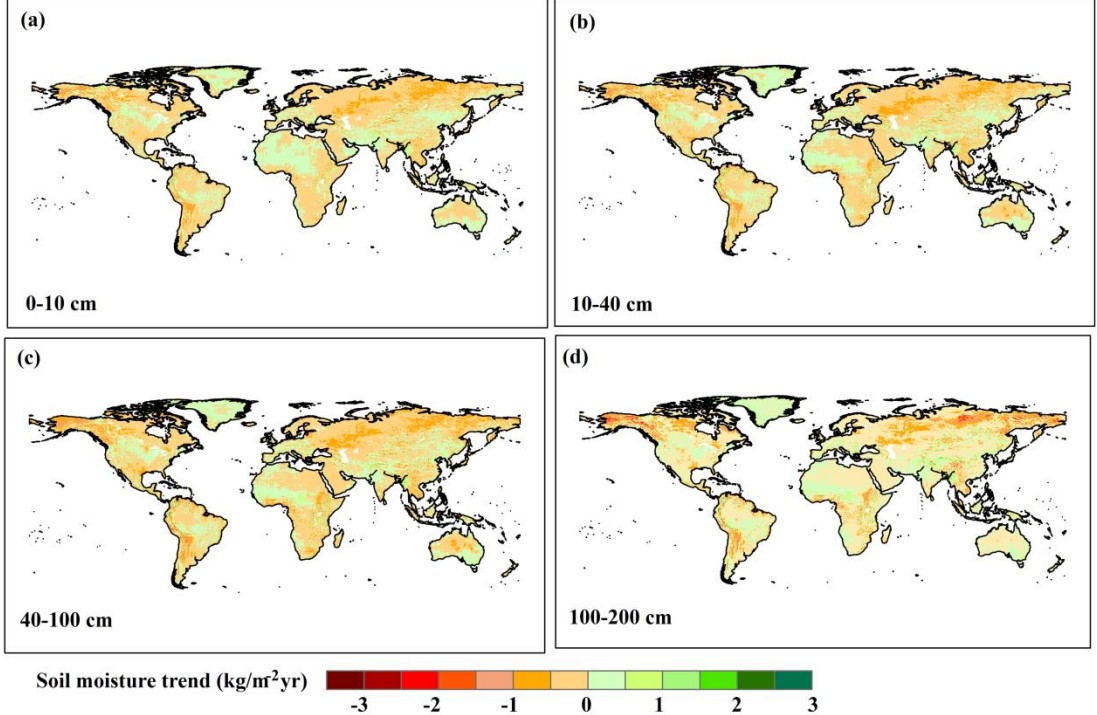


Fig. 5. Trends of soil moisture in different layers in 2000-2014.