# Peer review of "Dynamic changes of terrestrial net primary production and its"

_Hydrology and Earth System Sciences, 2016_

## Referee Comment (RC1) · Anonymous Referee #1 · 6 Apr 2016

This study tackles the association between NPP and ET under the climate change at a global scale, as well as inspects its regional variation. The topic is important and the findings are valuable. I recommend publishing after minor revision. Particularly, the presentation of the paper needs improvement.

Some major issues:

1. Throughout the paper the authors use the term "feedback" to refer to the "effect" of NPP on ET. I don't think "feedback" is the proper term here. "Feedback" means response. But I don't think the authors meant to say that ET first acts on NPP, and then NPP reacts on ET. Here "feedback" should be from NPP to climate. As the authors correctly stated in the paper, ET is the process that connects vegetation and climate, i.e., NPP can have feedback to climate through ET, but not have feedback to ET itself.

[Figure]

2. The authors found association/correlation between NPP and ET. However, correlation does not necessarily mean causal relationship. Among the three things: NPP, ET, and climate (temperature and precipitation), which one affects which one is a complex issue. The authors want to more clearly sort out the relationships among the three and be cautious on making strong statement that NPP is the cause for the change of ET.

3. Section 3.2: The first paragraph has a logic problem: If your goal is to understand the response of climate to NPP, why do you want to study the climate controlling factors on vegetation? Shouldn't it be reversed?

4. One of the conclusions of the paper is that if the effect of NPP on ET continues, regional droughts may get exacerbated. NPP and ET have a positive correlation, and authors found that the general trend globally is that NPP is increasing. Does this mean that more NPP globally will lead to more droughts? This sounds counterintuitive.

5. Did the authors calculate NPP and ET by themselves or simply use the MODIS products. This should be more clearly described in the paper. If MODIS products were used, then no algorithm details are needed; simply refer to relevant MODIS references. If the authors calculated those values by themselves, then more technical details are needed, including image selection, processing, and calculation processes; and in this case, the authors also need to properly cite the references for those algorithms and processes.

There are many language problems throughout the paper. I recommend that the authors find an English editor to proofread the paper. Below are some examples:

1. "Earth" should be "the Earth".

2. The phrase "under the content" appears in many places. I guess the authors meant to say "under the context". But in many places there should be better expressions.

3. I see "for the domain" in several places. I am not sure what it means.

4. "Temporal-spatial" should be "spatiotemporal".

5. The description of the NPP algorithm is hard to follow. If the authors decide to keep it, they should explain each symbol right below the equation where the symbol first appears.

6. Equation 8 is basic statistics; no need to include.

7. Line 20: should be "variable but positive trend".

8. Line 21: should be "which is consistent".

9. Line 22: "Trend" cannot be lower than "demand"; "deficit" cannot be "between"; what is "available water demand and supply for evapotranspiration"? This entire sentence needs to be rewritten.

10. Line 76: should be "came from".

11. Line 97: should be "allows the use".

12. Line 118: should be "the required MODIS data inputs for the ET algorithms".

13. Line 129: should be "t-Test".

14. Line 136: should be "While NPP in most part of NH increased, it decreased in most part of SH".

15. Line 156: should be "NPP and ET in SH have much higher variability in SH, where their association is less stronger than that in NH".

16. Line 159: "stronger variability" of what?

17. Line 161: should be "In places where the inter-annual . . ."

18. Line 164: should be "controlling factor for NPP variation".

19. Line 218: should be "proportional relationship".

20. Line 221: should be "over the past 15-year period".

21. Line 221: what is "P"? Also see comments 7 and 9 above.

22. Line 226: should be "affecting soil moisture".

23. Line 227: should be "temperature and solar radiation may cause soil moisture loss through evapotranspiration".

24. Line 231: Should be "Soil moisture is a common ecohydrologically confining factor".

---

## Referee Comment (RC2) · Anonymous Referee #2 · 16 Apr 2016

Review Report

Paper Title: Dynamic changes of terrestrial net primary production and its feedback to evapotranspiration

Authors: Zhi Li, Yaning Chen, Yang Wang, and Gonghuan Fang

Overall Comments:

In this paper, Li et al examined the change of terrestrial net primary production over the period of 2000 to 2014 based on NASA's Global 1-km MODIS NPP dataset and analyzed the factors determining the observed net primary productivity (NPP) change and its possible feedbacks to the atmosphere using statistical analysis. The major datasets used are all secondary by nature. The issues targeted at this research are

significant. The research design is generally justified, and the findings are interesting. I suggest that a moderate revision should be conducted to address or clarify several issues that are outlined below.

First, the research aimed to examine factors controlling net primary productivity (NPP) change and its feedbacks to the atmosphere, but the actual design was largely made to address the first part of the objective. More efforts on the second part would help bridge a knowledge gap and would definitely add values to the entire work.

Second, the authors need to be aware that from statistical perspective, correlation and causation are two related but distinct concepts. Correlation refers to how closely two sets of information or data are related, while causation means the act or process of causing that is often referred to as "cause and effect". Correlation may imply causation but in some cases, such a relation may not be true. Adding some more physical or theoretical explanation of the statistical relationship would help understand the causation.

Last, I understand English is not the mother language for these authors, and they must have made substantial efforts in polishing the English for the current submission. But there are still some small errors in English usage and grammar throughout the entire manuscript. These small errors collectively undermine the scholarly quality of this article. These authors are urged to find someone with competent English to edit the text.

Some more specific comments can be found from the next section of my review report.

Specific Comments:

1. Abstract

The abstract is too long and is not quite coherent. It needs to be better organized and written. Although the findings are present, it is not clear why these authors conducted this research, what would be the objectives, and what methods they have used.

2. Key words Should include a term for the geographical area

3. Introduction

The literature review could be more coherent with specific methods and critical issues (e.g. temporal and spatial scales) targeted. Also

Additional context would help justify the three specific objectives targeted.

4. Data and Methodology

Data: For each dataset, supply a brief description including the source. How did you resample and reclassify them?

Methods: The authors used the NPP datasets from MODIS, and it does not sethere is no need to have a lengthy discussion on the algorithms (Equations 1-7) used by NASA in a methodology section. This section should focus on the specific methods developed by these authors.

5. Results and Analyses

Statistical correlation and causation are related but different concepts. See my general comments. Need to check some sentences with strong statements (of causation).

6. Tables

Table 1: Line 446: remove "over". The heading for Column 4 is not correct. Please double check this.

7. Figures (changes in both captions and map legends)

Figure 1: use "Temporospatial". Which year of C and D for? Legends (C and D): remove "trend"

Figure 2: remove "trend" in the legend

Figure 3: add "or" in front of (d)

Figure 4: add a map to show NPP (should be (a)). Also remove "trend" in the legend.

Figure 5: Remove "trend" in the legend. Change the title into "—-in layers with different depths."

[Figure]

---

## Author Comment (AC1) · 1 May 2016

**Responses to the Referee's Comments**

**ID: HESS-2016-87-RC1**

We would like to thank the editor's decision regarding the revision of our manuscript. We are greatly thankful for the insightful and constructive comments from the anonymous reviewer. We have carefully studied them and revised the manuscript accordingly. This document contains our specific responses to the comments.

**Responses to Anonymous Referee #1's Comments:**

*1. Throughout the paper the authors use the term "feedback" to refer to the "effect"of NPP on ET. I don't think "feedback" is the proper term here. "Feedback" means response. But I don't think the authors meant to say that ET first acts on NPP, and then NPP reacts on ET. Here "feedback" should be from NPP to climate. As the authors correctly stated in the paper, ET is the process that connects vegetation and climate, i.e., NPP can have feedback to climate through ET, but not have feedback to ET itself.*

**Response**: Thanks for the reviewer's comment. We corrected the title as "Dynamic changes in terrestrial net primary production and their effects on evapotranspiration", also corrected it in the main text.

*2. The authors found association/correlation between NPP and ET. However, correlation does not necessarily mean causal relationship. Among the three things: NPP, ET, and climate (temperature and precipitation), which one affects which one is a complex issue. The authors want to more clearly sort out the relationships among the three and be cautious on making strong statement that NPP is the cause for the change of ET.*

**Response**: Table 1 and its description in the text are the correlations between NPP and four climatic variables (temperature, precipitation, net radiation and PDSI) for over both hemispheres. We have not made the quantitative association/correlation between NPP and ET. We just wanted to show that: Land actual evapotranspiration (ET) changes are positively with net primary production changes, especially in the Northern Hemisphere. Under the context of past warmest 15 years, the slightly increased inter-annual series of NPP and climate change promote decadal rises of global land ET. According to the reviewer's comment, we corrected several sentences

that seem strong statement that NPP is the cause for the change of ET. e.g. Abstract and 3.1 Spatiotemporal variation in global terrestrial NPP and its effects on ET.

**Abstract**. The Earth has experienced a dramatic increase in global climate warming since 2000, which has significantly influenced the global water cycle and vegetation activities. Despite these radical changes, there is little observational evidence to demonstrate the effects of vegetation variations on climate in different global geographical units. A few studies focused on feedback relating to the inter-annual variability of vegetation on the physical processes of atmosphere-land surfaces, especially with regards to evapotranspiration. Overall, the global inter-annual series of net primary production (NPP) slightly increased in 2000-2014 at a rate of 0.06 $PgC/yr^2$. More than 64% of vegetated land in the Northern Hemisphere (NH) showed increased NPP (at a rate of 0.13 $PgC/yr^2$), while 60.3% of vegetated land in the Southern Hemisphere (SH) showed a decreasing trend (at a rate of -0.18 $PgC/yr^2$). Temperature was the dominant control factor for vegetation growth in high latitudes in the NH, net radiation and precipitation were the main factors affecting NPP in the mid latitudes, and warming-associated large-scale drying trends led to decreases in NPP in the SH.

Vegetation productivity influences albedo and emissivity, both of which regulate evapotranspiration (ET). Changes in actual ET correlate positively with changes in NPP, especially in the NH, where increased vegetation productivity and climate change have promoted sharp rises in terrestrial ET (0.61 $mm/yr^2$). At the same time, anomalous dry conditions have caused a reduction in vegetation productivity and a near cessation of terrestrial ET growth in the SH (0.41 $mm/yr^2$). Although water vapor via vegetation transpiration can lead to increased regional atmospheric humidity over a short time span, it also preserves less water in the soil. This, in turn, accelerates reductions in soil moisture caused by warming and triggers negative feedback. Drought indices, along with precipitation-minus-evaporation calculations, point to an increased risk of drought in the 21st century.

**3.1 Spatiotemporal variations in global terrestrial NPP and their effects on ET**

The spatial patterns of global NPP from 2000 to 2014 showed a steadily decreasing trend from the equator to the Arctic and Antarctic (Fig. 1c). Overall, the inter-annual series of NPP increased moderately at a rate of 0.06 $PgC/yr^2$ over the past 15 years, and also shows different changes in the Northern and Southern Hemispheres. While

NPP in most parts of the NH increased (Fig. 1a), it decreased in most parts of the SH (Fig. 1b). Specifically, in the NH, 64% of vegetated land area experienced increased NPP, including large areas of North America, Western Europe, India, and eastern China. Regions with decreased NPP include Eastern Europe and higher latitudes of central and west Asia. In the SH, decreased NPP accounted for about 60.3% of vegetated land area, mainly concentrated in South America, South Africa, and western Australia. Furthermore, in the equatorial regions, Amazon rainforests had significantly decreased NPP, whereas African rainforests experienced an increasing trend (Fig. 1c). Because tropical rainforest NPP accounts for a large proportion of global NPP, decreases in SH NPP partially counteracted increases in NH NPP.

When we combined global LUCC characteristics, the results showed that shrubland has the greatest potential increasing trend of NPP (16.5 $gC/m^2/yr$) compared to other biomes, followed by grassland (12.5 $gC/m^2/yr$). This may be related to the expansion of woody vegetation over the past 15 years. In the Arctic tundra (Hughes et al., 2006) and lower latitudes in arid environments (Chen et al., 2014), experimental studies provided clear evidence that climate warming is sufficient to account for the expansion of shrubs and graminoids.

Changes in vegetation albedo and emissivity exert feedback on climate, which is especially obvious in ET (Field et al., 2007). The viability of vegetation cover can substantially modulate available surface energy and partition that energy into sensible and latent heat fluxes (Matsui et al., 2005). Vegetation changes can affect water and energy cycles in the land-atmosphere circulation by changing the physical characteristics of the land surface. For instance, the albedo of vegetation is less than that of bare soil, enabling it to absorb more energy and thus increase ET. Moreover, canopy height can change land surface roughness and affect the energy and momentum between land and air transport, and leaves promote ET through direct evaporation either from precipitation or transpiration interception from water uptake by roots.

Increased vegetation productivity and climate change may promote multi-decadal rises of global land ET (Zhang et al., 2015). Vegetation generally promotes land-atmosphere water exchange via transpiration through a biological process, changing soil moisture conditions and affecting the land-atmosphere feedback. The average mean of estimated global annual ET is 518.6 mm/yr, with an inter-annual trend of 0.46 $mm/yr^2$. Figures 1a and 1b show that the spatiotemporal changes of global ET are

consistent with NPP variations, especially in the NH. Furthermore, where their association is less than that in the NH, NPP and ET in the SH have much higher variability. The spatial inconsistency in the SH mainly occurred near the equator, e.g., southern African rainforests (Figs. 1b & 1d). These regions have high values of average annual precipitation and stronger variability of precipitation than elsewhere, causing greater changes to ET and its components (land-surface evaporation, canopy evaporation, and transpiration). In places where the inter-annual variability of NPP is small, the ET component of land-surface evaporation increases. In contrast, in areas with large inter-annual variability of NPP, such as shrubland and grass-dominant regions, the ET components of land-surface evaporation decline and transpiration increases.

*3. Section 3.2: The first paragraph has a logic problem: If your goal is to understand the response of climate to NPP, why do you want to study the climate controlling factors on vegetation? Shouldn't it be reversed?*

**Response**: Yes, this sentence is indeed reversed. "To understand why NPP variations in the Northern and Southern Hemispheres respond differently to different climates, we first estimated the spatial trends of climatic control factors and then analyzed the complex multiple climatic constraints to plant growth. "

*4. One of the conclusions of the paper is that if the effect of NPP on ET continues, regional droughts may get exacerbated. NPP and ET have a positive correlation, and authors found that the general trend globally is that NPP is increasing. Does this mean that more NPP globally will lead to more droughts? This sounds counterintuitive.*

**Response**: To more clearly clarify the conclusion, we added the following relevant statements:

**3.3 Continued effects of NPP on evapotranspiration likely to exacerbate regional droughts**

With respect to the impact of drought on the world's ecosystems, studies have been limited regarding the contribution of vegetation and terrestrial water cycle components to drought variations (Falloon et al., 2012; Teuling et al., 2013). However, the present lack of high-quality and long-term records of actual ET limits the forecasting of drought under climate change circumstances.

Drought indices are pointing to an increased risk of drought in the present century.

We used potential evapotranspiration (PET) as a surrogate measure of atmospheric moisture demand. PET is defined as the maximum quantity of water capable of being evaporated from soil and transpired from vegetation, whereas ET is the actual evaporation from water and soil, as well as transpiration from vegetation. Penman (1948) stated that ET had a proportional relationship with PET, and Bouchet (1963) hypothesized that a complementary feedback mechanism exists between ET and PET in water-limited regions. Overall, our investigation indicate that there is a proportional relationship between ET and PET in humid regions and a complementary one in arid regions (Fig. 1d and Fig. 4a). In PET, the combined impacts of temperature, solar radiation, vapor pressure and wind speed (Zhang et al., 2015) interact with NPP (Fig. 4b). Over the past 15 years, global PET showed an increasing trend of 1.72 mm/yr$^2$, while global P increased at a rate of 0.84 mm/yr$^2$. However, precipitation increases cannot offset evaporative demand, indicating a potential moisture deficit for water supplies constrained by ET. In other words, P is mostly being lost to ET rather than being allocated to other components of the energy and water cycles (Zhang et al., 2015).

Various factors, including vegetation, affect the intensity and spatial variation of drought. Vegetation generally promotes land-atmosphere water exchange via transpiration, changing soil moisture conditions and affecting the land-atmosphere feedback. Water vapor via transpiration can lead to increased regional atmospheric humidity over the short term, but preserve less water in the soil. This, in turn, accelerates reductions in soil moisture caused by warming and provides a negative feedback. Vegetation growth causes increased soil moisture evaporation, thus reducing the amount of soil water storage. Once vegetation suffers persistent drought, the vegetation biomass will rapidly decline and further intensify the drought. Thus, decreasing soil moisture tends to decrease net terrestrial radiation at the surface through increasing land-surface temperatures. If low levels of soil moisture persist for long enough, reductions in vegetation cover and vigor can occur. As land-surface temperatures rise, increases in precipitation are insufficient to offset increases in evaporative demand, indicating a potential moisture deficit for water supplies constrained by ET. This leads to soil water loss and reduced vegetation growth, along with higher risk of drought (Meng et al., 2014; Zhang et al., 2015). Soil moisture is an important sensor for measuring superficial wetness and dryness levels, and generally reflects the dryness and wetness of climate. Figure 5 illustrates the world-wide

decrease in soil moisture of four layers (0-10, 10-40, 40-100, and 100-200 cm). With precipitation being the most direct factor affecting soil moisture, temperature and solar radiation may cause soil moisture loss through ET. Available soil moisture is defined as the amount of water a plant can access in its root zone. Thus, spatial and temporal variations in soil moisture are closely related to vegetation growth (Davis and Pelsor, 2001; Yang et al., 2010).

*The added References:*

Falloon, P.D., Dankers, R., Betts, R.A., Jones, C.D., Booth, B.B.B. and Lambert, F.H.: Role of vegetation change in future climate under the A1B scenario and a climate stabilisation scenario, using the HadCM3C Earth system model. Biogeosciences, 9, 4739-4756, doi:10.5194/bg-9-4739-2012, 2012.

Meng, X.H., Evans, J.P. and McCabe, M.F.: The impact of observed vegetation changes on land–atmosphere feedbacks during drought. J. Hydrometeor, 15, 759-776, doi:10.1175/JHM-D-13-0130.1, 2014.

Teuling, A.J., van Loon, A.F., Seneviratne, S.I., Lehner, I., Aubinet, M., Heinesch, B., Bernhofer, C., Grünwald, T., Prasse, H. and Spank, U.: Evapotranspiration amplifies European summer drought, Geophys. Res. Lett., 40, 2071-2075, doi: 10.1002/grl.50495, 2013.

Zhang, K., Kimball, J.S., Nemani, R.R., Running, S.W., Hong, Y., Gourley, J.J. and Yu, Z.B.: Vegetation greening and climate change promote multidecadal rises of global land evapotranspiration, Sci. Rep, 5, 15956, doi: 10.1038/srep15956, 2015.

*5. Did the authors calculate NPP and ET by themselves or simply use the MODIS products. This should be more clearly described in the paper. If MODIS products were used, then no algorithm details are needed; simply refer to relevant MODIS references. If the authors calculated those values by themselves, then more technical details are needed, including image selection, processing, and calculation processes; and in this case, the authors also need to properly cite the references for those algorithms and processes.*

**Response**: We used the MODIS products and clarified them in the "2.1 Data", meanwhile deleted the algorithm details from "2.2 Methods".

**2.1 Data**

The monthly grid data of the temperature and precipitation series from 2000-2014, with a spatial resolution of 0.5 degrees, were collected from the Climatic Research Unit (http://www.cru.uea.ac.uk/data/). The radiation and soil moisture data series was issued by the Global Land Data Assimilation System (GLDAS-1), with a spatial resolution of 0.25 degrees (http://gdata1.sci.gsfc.nasa.gov/daac-bin/

G3/gui.cgi?instance_id=GLDAS025_M). The depths of the four soil layers are: 0-10 cm, 10-40 cm, 40-100 cm, and 100-200 cm. The quality of the GLDAS data set was assessed against available observations from multiple sources (Zhang et al., 2008; Chen et al., 2013).

The monthly data of the Palmer Drought Severity Index (PDSI), with a spatial resolution of 2.5 degrees, was available at http://www.cgd.ucar.edu/cas/catalog/climind/pdsi.html. As an indicator of land-surface moisture conditions, PDSI has been widely used for the routine monitoring and assessment of global and regional drought conditions. The global dry areas were defined as PDSI < -3.0, while the wet areas were defined as PDSI > + 3.0 (Dai et al., 2004).

We used the Global Land Cover Characterization data from the International Geosphere-Biosphere Program (IGBP) in 2000 (http://edc2.usgs.gov/glcc/glcc.php), along with MODIS in 2000 and 2013 (http://modis.gsfc.nasa.gov/data/dataprod/mod12.php). From these data, a routinely integrated classification of land use/cover change (LUCC) characteristics was obtained based on the feature fusion processes.

The global 1-km MODIS NPP datasets (2000-2014) are from MOD17. NPP estimations are typically model-based and biogeochemical, and are generated from a larger set of simulated C fluxes between the atmosphere and terrestrial ecosystems (Ito et al., 2011). A better agreement of MODIS and terrestrial NPP estimates allows the use of MODIS in large-scale estimates (Neumann et al., 2015).

The MODIS evapotranspiration datasets (2000-2014) from MOD16 are estimated using Mu and colleagues' (2011) improved ET algorithm over Mu et al.'s (2007) previous paper. Based on the energy-balance theory and the Penman-Monteith equation, the required MODIS data inputs for the ET algorithms include daily meteorology (temperature, actual vapor pressure, and incoming solar radiation) remotely-sensed land cover, FPAR/LAI, and albedo (Friedl et al., 2010; Myneni et al., 2002). We unified the spatiotemporal resolution of these data from different sources, based on re-sampling (nearest neighbor interpolation) and re-classification techniques.

**Specific minor comments**:

*1. "Earth" should be "the Earth".*

**Response**: We corrected it in the revision.

*2. The phrase "under the content" appears in many places. I guess the authors meant to say "under the context". But in many places there should be better expressions.*

**Response**: We corrected it in the revision.

*3. I see "for the domain" in several places. I am not sure what it means.*

**Response**: "for the domain" that we used means "in the same region". Maybe it is inapposite, so that we have deleted these words in the revision.

*4. "Temporal-spatial" should be "spatiotemporal".*

**Response**: We corrected it in the revision.

*5. The description of the NPP algorithm is hard to follow. If the authors decide to keep it, they should explain each symbol right below the equation where the symbol first appears.*

**Response**: According to the major comment 5, we described the NPP products in the "2.1 Data", meanwhile deleted the algorithm details from "2.2 Methods".

*6. Equation 8 is basic statistics; no need to include.*

**Response**: We have deleted the Equation 8 according to the reviewer's comment.

*7. Line 20: should be "variable but positive trend".*
*8. Line 21: should be "which is consistent".*

**Response**: We corrected them in the revision.

*9. Line 22: "Trend" cannot be lower than "demand"; "deficit" cannot be "between"; what is "available water demand and supply for evapotranspiration"? This entire sentence needs to be rewritten.*

**Response**: We have rewritten this entire sentence: "But precipitation increases cannot offset evaporative demand, indicating a potential moisture deficit for water supplies constrained by evapotranspiration. This leads to soil water loss and reduced"

*10. Line 76: should be "came from".*
*11. Line 97: should be "allows the use".*
*12. Line 118: should be "the required MODIS data inputs for the ET algorithms".*

*13. Line 129: should be "t-Test".*

*14. Line 136: should be "While NPP in most part of NH increased, it decreased in most part of SH".*

**Response**: Thanks for the reviewer's comments. We corrected the specific minor comments of 10-14 in the revision, respectively.

*15. Line 156: should be "NPP and ET in SH have much higher variability in SH, where their association is less stronger than that in NH".*

**Response**: We have rewritten this entire sentence as follows: "NPP and ET in SH have much higher variability in SH, where their association is less than that in NH".

*16. Line 159: "stronger variability" of what?*

**Response**: These regions have high values of average annual precipitation and stronger variability of precipitation than elsewhere.

*17. Line 161: should be "In places where the inter-annual: : :"*

*18. Line 164: should be "controlling factor for NPP variation".*

*19. Line 218: should be "proportional relationship".*

*20. Line 221: should be "over the past 15-year period".*

**Response**: Thanks for the reviewer's comments. We corrected the specific minor comments of 17-20 in the revision, respectively.

*21. Line 221: what is "P"? Also see comments 7 and 9 above.*

**Response**: We have rewritten this entire sentence as follows: "Global PET showed an increasing trend of 1.72 mm/yr$^2$ over the past 15-year period, while global P increased at a rate of 0.84 mm/yr$^2$."

*22. Line 226: should be "affecting soil moisture".*

*23. Line 227: should be "temperature and solar radiation may cause soil moisture loss through evapotranspiration".*

*24. Line 231: Should be "Soil moisture is a common ecohydrologically confining factor".*

**Response**: Thanks for the reviewer's comments. We corrected the specific minor comments of 22-24 in the revision, respectively.

---

## Author Comment (AC2) · 1 May 2016

Responses to the Referee's Comments
peer review response
en

**Responses to the Referee's Comments**

**ID: HESS-2016-87-RC2**

We would like to thank the editor's decision regarding the revision of our manuscript. We are greatly thankful for the insightful and constructive comments from the anonymous reviewer. We have carefully studied them and revised the manuscript accordingly. This document contains our specific responses to the comments.

**Responses to Anonymous Referee #2's Comments:**

*1. First, the research aimed to examine factors controlling net primary productivity (NPP) change and its feedbacks to the atmosphere, but the actual design was largely made to address the first part of the objective. More efforts on the second part would help bridge a knowledge gap and would definitely add values to the entire work.*

**Response**: We rewrote the 3.1 & 3.3 parts according to the reviewer's comment.

**3.1 Spatiotemporal variations in global terrestrial NPP and their effects on ET**

The spatial patterns of global NPP from 2000 to 2014 showed a steadily decreasing trend from the equator to the Arctic and Antarctic (Fig. 1c). Overall, the inter-annual series of NPP increased moderately at a rate of 0.06 $PgC/yr^2$ over the past 15 years, and also shows different changes in the Northern and Southern Hemispheres. While NPP in most parts of the NH increased (Fig. 1a), it decreased in most parts of the SH (Fig. 1b). Specifically, in the NH, 64% of vegetated land area experienced increased NPP, including large areas of North America, Western Europe, India, and eastern China. Regions with decreased NPP include Eastern Europe and higher latitudes of central and west Asia. In the SH, decreased NPP accounted for about 60.3% of vegetated land area, mainly concentrated in South America, South Africa, and western Australia. Furthermore, in the equatorial regions, Amazon rainforests had significantly decreased NPP, whereas African rainforests experienced an increasing trend (Fig. 1c). Because tropical rainforest NPP accounts for a large proportion of global NPP, decreases in SH NPP partially counteracted increases in NH NPP.

When we combined global LUCC characteristics, the results showed that shrubland has the greatest potential increasing trend of NPP (16.5 $gC/m^2/yr$) compared to other biomes, followed by grassland (12.5 $gC/m^2/yr$). This may be related to the expansion of woody vegetation over the past 15 years. In the Arctic tundra (Hughes et al., 2006)

and lower latitudes in arid environments (Chen et al., 2014), experimental studies provided clear evidence that climate warming is sufficient to account for the expansion of shrubs and graminoids.

Changes in vegetation albedo and emissivity exert feedback on climate, which is especially obvious in ET (Field et al., 2007). The viability of vegetation cover can substantially modulate available surface energy and partition that energy into sensible and latent heat fluxes (Matsui et al., 2005). Vegetation changes can affect water and energy cycles in the land-atmosphere circulation by changing the physical characteristics of the land surface. For instance, the albedo of vegetation is less than that of bare soil, enabling it to absorb more energy and thus increase ET. Moreover, canopy height can change land surface roughness and affect the energy and momentum between land and air transport, and leaves promote ET through direct evaporation either from precipitation or transpiration interception from water uptake by roots.

Increased vegetation productivity and climate change may promote multi-decadal rises of global land ET (Zhang et al., 2015). Vegetation generally promotes land-atmosphere water exchange via transpiration through a biological process, changing soil moisture conditions and affecting the land-atmosphere feedback. The average mean of estimated global annual ET is 518.6 mm/yr, with an inter-annual trend of 0.46 mm/yr$^2$. Figures 1a and 1b show that the spatiotemporal changes of global ET are consistent with NPP variations, especially in the NH. Furthermore, where their association is less than that in the NH, NPP and ET in the SH have much higher variability. The spatial inconsistency in the SH mainly occurred near the equator, e.g., southern African rainforests (Figs. 1b & 1d). These regions have high values of average annual precipitation and stronger variability of precipitation than elsewhere, causing greater changes to ET and its components (land-surface evaporation, canopy evaporation, and transpiration). In places where the inter-annual variability of NPP is small, the ET component of land-surface evaporation increases. In contrast, in areas with large inter-annual variability of NPP, such as shrubland and grass-dominant regions, the ET components of land-surface evaporation decline and transpiration increases.

**3.3 Continued effects of NPP on evapotranspiration likely to exacerbate regional droughts**

With respect to the impact of drought on the world's ecosystems, studies have been limited regarding the contribution of vegetation and terrestrial water cycle components to drought variations (Falloon et al., 2012; Teuling et al., 2013). However, the present lack of high-quality and long-term records of actual ET limits the forecasting of drought under climate change circumstances.

Drought indices are pointing to an increased risk of drought in the present century. We used potential evapotranspiration (PET) as a surrogate measure of atmospheric moisture demand. PET is defined as the maximum quantity of water capable of being evaporated from soil and transpired from vegetation, whereas ET is the actual evaporation from water and soil, as well as transpiration from vegetation. Penman (1948) stated that ET had a proportional relationship with PET, and Bouchet (1963) hypothesized that a complementary feedback mechanism exists between ET and PET in water-limited regions. Overall, our investigation indicate that there is a proportional relationship between ET and PET in humid regions and a complementary one in arid regions (Fig. 1d and Fig. 4a). In PET, the combined impacts of temperature, solar radiation, vapor pressure and wind speed (Zhang et al., 2015) interact with NPP (Fig. 4b). Over the past 15 years, global PET showed an increasing trend of 1.72 $mm/yr^2$, while global P increased at a rate of 0.84 $mm/yr^2$. However, precipitation increases cannot offset evaporative demand, indicating a potential moisture deficit for water supplies constrained by ET. In other words, P is mostly being lost to ET rather than being allocated to other components of the energy and water cycles (Zhang et al., 2015).

Various factors, including vegetation, affect the intensity and spatial variation of drought. Vegetation generally promotes land-atmosphere water exchange via transpiration, changing soil moisture conditions and affecting the land-atmosphere feedback. Water vapor via transpiration can lead to increased regional atmospheric humidity over the short term, but preserve less water in the soil. This, in turn, accelerates reductions in soil moisture caused by warming and provides a negative feedback. Vegetation growth causes increased soil moisture evaporation, thus reducing the amount of soil water storage. Once vegetation suffers persistent drought, the vegetation biomass will rapidly decline and further intensify the drought. Thus, decreasing soil moisture tends to decrease net terrestrial radiation at the surface through increasing land-surface temperatures. If low levels of soil moisture persist for long enough, reductions in vegetation cover and vigor can occur. As land-surface

temperatures rise, increases in precipitation are insufficient to offset increases in evaporative demand, indicating a potential moisture deficit for water supplies constrained by ET. This leads to soil water loss and reduced vegetation growth, along with higher risk of drought (Meng et al., 2014; Zhang et al., 2015). Soil moisture is an important sensor for measuring superficial wetness and dryness levels, and generally reflects the dryness and wetness of climate. Figure 5 illustrates the world-wide decrease in soil moisture of four layers (0-10, 10-40, 40-100, and 100-200 cm). With precipitation being the most direct factor affecting soil moisture, temperature and solar radiation may cause soil moisture loss through ET. Available soil moisture is defined as the amount of water a plant can access in its root zone. Thus, spatial and temporal variations in soil moisture are closely related to vegetation growth (Davis and Pelsor, 2001; Yang et al., 2010).

*2. Second, the authors need to be aware that from statistical perspective, correlation and causation are two related but distinct concepts. Correlation refers to how closely two sets of information or data are related, while causation means the act or process of causing that is often referred to as "cause and effect". Correlation may imply causation but in some cases, such a relation may not be true. Adding some more physical or the-oretical explanation of the statistical relationship would help understand the causation.*

**Response**: Table 1 and its description in the text are the correlations between NPP and four climatic variables (temperature, precipitation, net radiation and PDSI) for over both hemispheres. We have not made the quantitative association/correlation between NPP and ET. We just wanted to show that: Land actual evapotranspiration (ET) changes are positively with net primary production changes, especially in the Northern Hemisphere. Under the context of past warmest 15 years, the slightly increased inter-annual series of NPP and climate change promote decadal rises of global land ET. According to the reviewer's comment, we corrected several sentences that seem strong statement that NPP is the cause for the change of ET. Meanwhile, we added some more physical or theoretical explanation.

e.g. 3.1 & 3.3 (Please see the response of comment 1)

**3.2 Controlling factors for NPP variations**

Water, temperature, and radiation interact to impose complex and varying limitations

on vegetation activities in different parts of the world. To understand why NPP variations in the Northern and Southern Hemispheres respond differently to different climates, we first estimated the spatial trends of climatic control factors and then analyzed the complex multiple climatic constraints to plant growth. A comprehensive interpretation of interactive climatic controls on plant productivity showed that water, temperature, and radiation are the key factors affecting vegetation growth. Globally, growth was most strongly limited by water availability on 40% of the Earth's vegetated surface, while temperature limitations exerted the main controlling influence on 33% of the surface, and radiation on 27% (Nemani et al., 2003).

From 2000 to 2014, overall trends of average annual temperature (0.007 ℃/yr$^2$) and precipitation (0.84 mm/yr$^2$) experienced world-wide increases while showing different temporal change patterns (Figs. 2a & 2b). Eastern Europe, South America, southern Africa and western Australia experienced warming combined with decreased precipitation (warm-dry trend), whereas southeast North America, western Europe, east Russia, and African rainforests experienced warming combined with increased precipitation (warm-wet trend). Meanwhile, net radiation (Rn) increased in the equatorial tropics and arid regions in Northwestern China, but decreased in the Arctic and Antarctic (Fig. 2c). The Palmer Drought Severity Index (PDSI) is a widely-used index that correlates with soil moisture during warm seasons (Dai et al., 2004; 2013). Generally, a lower PDSI implies a drier climate. Global PDSI decreased at a rate of 0.04/yr$^2$ from 2000 to 2014, suggesting an increased risk of drought in the 21st century. The spatial trend of PDSI shows that the eastern and northern coasts of North America, along with the African continent, Eurasia and southern South America, exhibited obvious drought trends in 2000-2014. Northern China, parts of Mongolia, and western Russia near Lake Baikal also experienced a drying trend (Fig. 2d). Warming-induced drying resulted from increased ET and was most prevalent over mid-high latitudes in the NH. Drought develops with periods of low-accumulated precipitation and is exacerbated by high temperatures.

We analyzed partial correlations between NPP and temperature (T), precipitation (P), Net radiation (Rn) and PDSI during growing seasons to determine their respective contributions across different regions (Fig. 3, Table 1). The climate changes during 2000-2014 have made precipitation, temperature, and radiation somewhat beneficial to plant growth. In the NH, climate changes have eased multiple climatic constraints

to plant growth through earlier springs, but the continuous warming may offset these benefits.

In high latitudes of the NH (>47.5 °N), temperature has a positive correlation with NPP (R=0.6). Significant warming generally lengthens growing seasons for vegetation and promotes plant growth in tundra regions, so the recent warming in this region has increased NPP (Fig. 3a). For northern mid and low latitudes (<47.5 °N), where large areas are classified as having an arid climate, vegetation is short-rooted. NPP has a significant negative correlation with P (R=0.7, p<0.05) (Fig. 3b) and is also correlated to Rn (Fig. 3c). In areas of high elevation such as the Tibetan plateau (which is similar to high latitudes), temperature is the dominant control factor in vegetation growth.

Equatorial Amazon rainforests experienced significantly decreased NPP, whereas African rainforests exhibited an increasing trend. These changes in equatorial regions are mainly related to the warming in the Amazon along with increasing precipitation in African rainforests. High temperatures caused higher rates of ET, generally reducing soil water availability for vegetation in the Amazon.

In the SH, we noted a significant correlation (r =0.7, p<0.05) between NPP and PDSI (Fig. 3d). The warming trend induced a much higher evaporative demand and led to a drying trend, except for the afore-mentioned increased precipitation in African rainforests. The PDSI in African rainforests also showed a slight increasing trend. A high T-value can increase both vapour pressure deficiency and moisture deficit and lead to a dryer environment. The general drought event across the SH, which was induced by extreme heat and a precipitation deficit, has resulted in a net water availability reduction, ultimately reducing NPP. Hence, warming-associated drying directly caused the significant decreasing trend of NPP in the SH.

*3. Last, I understand English is not the mother language for these authors, and they must have made substantial efforts in polishing the English for the current submission. But there are still some small errors in English usage and grammar throughout the entire manuscript. These small errors collectively undermine the scholarly quality of this article. These authors are urged to find someone with competent English to edit the text.*

**Response**: We carefully modified the grammar and sentence structure with the assistance of native English-speaking editors.

*4. The abstract is too long and is not quite coherent. It needs to be better organized and written. Although the findings are present, it is not clear why these authors conducted this research, what would be the objectives, and what methods they have used.*

**Response**: We have rewritten the abstract.

**Abstract**. The Earth has experienced a dramatic increase in global climate warming since 2000, which has significantly influenced the global water cycle and vegetation activities. Despite these radical changes, there is little observational evidence to demonstrate the effects of vegetation variations on climate in different global geographical units. A few studies focused on feedback relating to the inter-annual variability of vegetation on the physical processes of atmosphere-land surfaces, especially with regards to evapotranspiration. Overall, the global inter-annual series of net primary production (NPP) slightly increased in 2000-2014 at a rate of 0.06 $PgC/yr^2$. More than 64% of vegetated land in the Northern Hemisphere (NH) showed increased NPP (at a rate of 0.13 $PgC/yr^2$), while 60.3% of vegetated land in the Southern Hemisphere (SH) showed a decreasing trend (at a rate of -0.18 $PgC/yr^2$). Temperature was the dominant control factor for vegetation growth in high latitudes in the NH, net radiation and precipitation were the main factors affecting NPP in the mid latitudes, and warming-associated large-scale drying trends led to decreases in NPP in the SH.

Vegetation productivity influences albedo and emissivity, both of which regulate evapotranspiration (ET). Changes in actual ET correlate positively with changes in NPP, especially in the NH, where increased vegetation productivity and climate change have promoted sharp rises in terrestrial ET (0.61 $mm/yr^2$). At the same time, anomalous dry conditions have caused a reduction in vegetation productivity and a near cessation of terrestrial ET growth in the SH (0.41 $mm/yr^2$). Although water vapor via vegetation transpiration can lead to increased regional atmospheric humidity over a short time span, it also preserves less water in the soil. This, in turn, accelerates reductions in soil moisture caused by warming and triggers negative feedback. Drought indices, along with precipitation-minus-evaporation calculations, point to an increased risk of drought in the 21st century.

*5. Key words should include a term for the geographical area*

**Response**: Journal of *Hydrol. Earth Syst. Sci.* looks probably no keywords.

*6. Introduction: The literature review could be more coherent with specific methods and critical issues (e.g. temporal and spatial scales) targeted. Also additional context would help justify the three specific objectives targeted.*

**Response**: We have rewritten the Introduction.

**Introduction**

Organizations such as the Intergovernmental Panel on Climate Change (IPCC) and the World Meteorological Organization (WMO) have reported that the past decade was the warmest on record. Global warming indicates a general acceleration or intensification of the global hydrological cycle and thus an alteration in the process of evapotranspiration (ET) (Wentz et al., 2007; Douville et al., 2013), with implications for the response and mutual feedback of ecosystem services (Jung et al., 2010; Davie et al., 2013). Most of the recent research in this area has analyzed the impacts of climate change on vegetation activities, although some of the research on the feedback of terrestrial ecosystems to climate change has centered on the ecosystems' potential role as carbon sources (Field et al., 2007). Only a few studies focused on feedback relating to the inter-annual variability of vegetation on the physical processes of land surfaces and climate systems (Zhi et al., 2009), especially with regards to ET.

Terrestrial net primary production (NPP) can be defined as the amount of photosynthetically fixed carbon available to the first heterotrophic level in an ecosystem, and links terrestrial biota with atmospheric systems (Beer et al., 2010). Numerous ecosystem models have attempted to estimate terrestrial NPP, owing to its importance for ecological and social systems, both regionally and globally (Chen et al., 2012; Potter et al., 2012; Pan et al., 2014). From 1982 to 1999, climatic changes enhanced plant growth globally, especially in the northern mid and high latitudes (Nemani, 2003); this was followed, in 2000-2009, by a drought-induced reduction in global NPP (Zhao and Running, 2010). Gang et al. (2015) projected the dynamics of NPP in response to future anticipated climate changes in the 2030s, 2050s and 2070s, and found that global NPP would show an increasing trend. In particular, NPP at high latitudes in the Northern Hemisphere (NH) would likely be more sensitive to future climate change. The interaction of water, temperature, and radiation has imposed complex and varying limitations on vegetation activities in different regions of the world. However, clear data on spatiotemporal variations and attributes in global terrestrial NPP within the context of high variability warming are still lacking.

Meanwhile, several studies have indicated that climate constraints (e.g., increasing temperatures and solar radiation) are relaxing (Nemani et al., 2003).

A recent study suggested that vegetation productivity influences albedo and emissivity, which then strongly regulate global climate (Chapin et al., 2011). This is especially obvious in evapotranspiration (ET), which is a central process in the climate system and a nexus of the water, energy and carbon cycles. Hence, it plays a pivotal role in maintaining the water and heat balance. Shen and colleagues (2015) reported that, in contrast to the Arctic region (i.e., positive feedback to warming), increased vegetation activity may attenuate daytime warming by enhancing actual ET as a cooling process on the Tibetan Plateau. Zhang et al. (2015) investigated how climate change and recent vegetation greening promote multi-decadal rises of global ET, while an anomalous drought between 2000 and 2009 led to reduced vegetation productivity in the Southern Hemisphere (SH) (Zhao and Running, 2010). However, little observational evidence exists to demonstrate vegetation feedback on climate across different global geographical units.

Investigating factors that control changes in NPP and its feedback effects could provide important clues to the underlying mechanisms and complex interactions between ecosystems and climate systems (Tian et al., 2000; 2012). Having a clear understanding of the land's biophysical feedback to the atmosphere is crucial if we are to simulate regional climate accurately. In our study, we investigated the following three major points of interest: 1) whether the high volatility temperature of the past decade continued to increase NPP, or if different climate constraints were at play; 2) why NPP variations in the Northern and Southern Hemispheres respond differently to climate changes; and 3) what the spatiotemporal variation of NPP is, and what its effects are on ET.

*7. Data and Methodology*

*Data: For each dataset, supply a brief description including the source. How did you resample and reclassify them? Methods: The authors used the NPP datasets from MODIS, and it does not set here is no need to have a lengthy discussion on the algorithms (Equations 1-7) used by NASA in a methodology section. This section should focus on the specific methods developed by these authors.*

**Response**: We clarified the source for each dataset in the "2.1 Data", meanwhile deleted the unnecessary algorithm details from "2.2 Methods".

**2.1 Data**

The monthly grid data of the temperature and precipitation series from 2000-2014, with a spatial resolution of 0.5 degrees, were collected from the Climatic Research Unit (http://www.cru.uea.ac.uk/data/). The radiation and soil moisture data series was issued by the Global Land Data Assimilation System (GLDAS-1), with a spatial resolution of 0.25 degrees (http://gdata1.sci.gsfc.nasa.gov/daac-bin/G3/gui.cgi?instance_id=GLDAS025_M). The depths of the four soil layers are: 0-10 cm, 10-40 cm, 40-100 cm, and 100-200 cm. The quality of the GLDAS data set was assessed against available observations from multiple sources (Zhang et al., 2008; Chen et al., 2013).

The monthly data of the Palmer Drought Severity Index (PDSI), with a spatial resolution of 2.5 degrees, was available at http://www.cgd.ucar.edu/cas/catalog/climind/pdsi.html. As an indicator of land-surface moisture conditions, PDSI has been widely used for the routine monitoring and assessment of global and regional drought conditions. The global dry areas were defined as PDSI < -3.0, while the wet areas were defined as PDSI > + 3.0 (Dai et al., 2004).

We used the Global Land Cover Characterization data from the International Geosphere-Biosphere Program (IGBP) in 2000 (http://edc2.usgs.gov/glcc/glcc.php), along with MODIS in 2000 and 2013 (http://modis.gsfc.nasa.gov/data/dataprod/mod12.php). From these data, a routinely integrated classification of land use/cover change (LUCC) characteristics was obtained based on the feature fusion processes.

The global 1-km MODIS NPP datasets (2000-2014) are from MOD17. NPP estimations are typically model-based and biogeochemical, and are generated from a larger set of simulated C fluxes between the atmosphere and terrestrial ecosystems (Ito et al., 2011). A better agreement of MODIS and terrestrial NPP estimates allows the use of MODIS in large-scale estimates (Neumann et al., 2015).

The MODIS evapotranspiration datasets (2000-2014) from MOD16 are estimated using Mu and colleagues' (2011) improved ET algorithm over Mu et al.'s (2007) previous paper. Based on the energy-balance theory and the Penman-Monteith equation, the required MODIS data inputs for the ET algorithms include daily meteorology (temperature, actual vapor pressure, and incoming solar radiation) remotely-sensed land cover, FPAR/LAI, and albedo (Friedl et al., 2010; Myneni et al., 2002). We unified the spatiotemporal resolution of these data from different sources, based on re-sampling (nearest neighbor interpolation) and re-classification techniques.

**2.2 Methods**

*Trend analysis.* To further discern the trends of yearly NPP and ET, we examined linear trend estimations on a per-pixel basis to establish a linear regression relationship between variables and time.

*Partial correlation analysis.* This method is used to describe the relationship between two variables while taking away the effects of several other variables. The partial correlation of $x_1$ and $x_2$ is adjusted for a third variable y (at a significance level of 0.05 by t-Test):

$$r_{x1x2 \cdot y} = \frac{r_{x1x2} - r_{x1y}r_{x2y}}{\sqrt{\left(1 - r_{x1y}^2\right)\left(1 - r_{x2y}^2\right)}} \tag{1}$$

*8. Results and Analyses*

*Statistical correlation and causation are related but different concepts. See my general comments. Need to check some sentences with strong statements (of causation).*

**Response**: Thanks for the reviewer's comment. Please see the response to the comment 2.

*9. Table 1: remove "over". The heading for Column 4 is not correct. Please double check this.*

**Response**: We corrected the table 1 in the revision.

Table 1. Correlations between NPP and climatic variables for both hemispheres

| Zones | NPP trend PgC yr$^{-2}$ | T trend ℃ yr$^{-2}$ | P trend mm yr$^{-2}$ | Rn trend W m$^{-2}$ yr$^{-2}$ | PDSI trend yr$^{-2}$ |
|---|---|---|---|---|---|
| NH high latitudes (>47.5 N) | y=0.02x+30.51 | y=0.021x-5.75 **R= 0.60*** | y=0.104x+46.58 R=0.29 | y=-5.21x+453.6 R=0.45 | y=-0.005x+0.23 R=0.44 |
| NH mid/low latitudes (<47.5 N) | y=0.07x+45.68 | y=0.009x+18.3 R= -0.17 | y=0.341x+76.8 **R= 0.70**** | y=3.239x+105.9 R=0.50 | y=0.006x-0.46 R=0.56 |
| South Hemisphere | y=-0.18x+78.37 | y=0.010x+21.6 R= -0.53 | y=0.074x+116.8 R=0.37 | y=2.455x+129.4 R=0.43 | y=-0.042x+0.33 **R= 0.70**** |

*10. Figures (changes in both captions and map legends)*

*Figure 1: use "Temporospatial". Which year of C and D for? Legends (C and D): remove "trend"*

**Response**: We updated Figure 1 according to the reviewer's comments.

[Figure]

Fig. 1. Temporospatial variations in global terrestrial NPP and ET from 2000-2014. (a) Inter-annual variations of NPP and ET in the Northern Hemisphere (NH). (b) Inter-annual variations of NPP and ET in the Southern Hemisphere (SH). (c) Spatial pattern of NPP trend from 2000-2014. (d) Spatial pattern of ET trend from 2000-2014.

*11. Figure 2: remove "trend" in the legend*

**Response**: We updated Figure 2 according to the reviewer's comments.

[Figure]

Fig. 2. Trends of air temperature (T), precipitation (P), net radiation (Rn) and Palmer Drought Severity Index (PDSI) from 2000-2014.

*12. Figure 3: add "or" in front of (d)*

**Response**: We corrected the caption according to the reviewer's comments.

Fig. 3. Partial correlations between NPP and (a) Temperature, (b) Precipitation, (c) Net radiation, or (d) PDSI in growing season

*Figure 4: add a map to show NPP (should be (a)). Also remove "trend" in the legend.*

**Response**: We have removed "trend" in the legend. Because of the NPP has already showed in the figure 1(c), here probably does not need to appear again.

[Figure]

Fig. 4. (a) Spatial pattern of PET trend. (b) Partial correlations between NPP and PET.

*Figure 5: Remove "trend" in the legend. Change the title into "—-in layers with different depths."*

**Response**: We updated the Figure 5 according to the reviewer's comments.

[Figure]

Fig. 5. Trends of soil moisture -in layers with different depths in 2000-2014.

---

## Referee Comment (RC3) · Anonymous Referee #1 · 4 May 2016

I still strongly recommend that the authors find an English editor who is experienced with scientific writing to proofread this manuscript. Only grammatical checking is not enough. For example, the every first sentence of the Abstract is problematic, "The Earth has experienced a dramatic increase in global climate warming since 2000". People may interpret sentence as that the Earth has been increasing (getting bigger and bigger) since 2000, which apparently is not what the authors intend to say.

The first couple of sentences of the abstract can be revised as follows, "The dramatic increase of global temperature since year 2000 has a considerable impact on the global water cycle and vegetation dynamics, which has been extensively studied. However, little has been done about recent feedback of vegetation to climate in different parts of the world."

[Figure]

Based on this first sentence, it seems that the main research question of this study is about this vegetation (NPP) -> climate feedback, but then it is not clear what purpose the next sentence "A few studies focused on feedback . . . evapotranspiration" is serving. First, if there are already "a few studies", then what is special (novel) with your study? Do you actually mean "few" instead of "a few"? Second, what is the role of evapotranspiration in addressing NPP -> climate feedback? You need to provide a connection between it and the main research question (NPP -> climate).

The next couple of sentences reporting NPP changes in north and south hemispheres are another jump, without apparent connection with the NPP->climate feedback. The last sentence of the first paragraph of Abstract entirely reverses the question: now you are talking about climate -> NPP.

It is uncommon that an abstract has more than one paragraph. More importantly, you need a sentence or two to connect the content of the second paragraph to the main issue of this study (NPP -> climate). I think the main point you want to make is that vegetation has a feedback to climate through ET.

I think the statements (predictions) you made in the second paragraph are confusing. It seems to me that what you are saying is that no matter NPP increases (in NH) or decreases (in SH), it will lead to drought. I don't see sustentative analysis in your paper that supports these predictions. For example, in Figure 3, NPP and PDSI are positively correlated in some places and negatively correlated in other places. I don't see that you have an analysis to show which (positive or negative) is dominant in different regions.

In the last sentence of Abstract, you use a term "negative feedback". I am not sure what you mean by this term. In system science, "negative feedback" means that the feedbacks of the two parties "discourage" each other and finally the system reaches an equilibrium status. "Positive feedback" means that the feedbacks of the two parties "encourage" each other and "the flapping of the wings of a butterfly" may finally cause "a big storm".

My main suggestion is that the authors want to first clearly sort out the main logic of this paper. If the main research question is the feedback of NPP to climate, and the main point is that EP is the means of this feedback, then this logic needs to be very clearly presented and all the materials need to be organized around it. Without this big picture in mind, it is easy to get lost in details, e.g., flip back and forth between NPP -> climate and climate -> NPP.

---

## Author Comment (AC4) · 13 May 2016

**Responses to the Referee's Comments**

**ID: HESS-2016-87-RC3**

We are greatly thankful for the constructive comments from the anonymous reviewer. We have carefully studied them and revised the manuscript accordingly. This document contains our specific responses to the comments.

**Responses to Anonymous Referee #1's Comments:**

*1. Based on this first sentence, it seems that the main research question of this study is about this vegetation (NPP) -> climate feedback, but then it is not clear what purpose the next sentence "A few studies focused on feedback evapotranspiration is serving. First, if there are already "a few studies", then what is special (novel) with your study? Do you actually mean "few" instead of "a few"? Second, what is the role of evapotranspiration in addressing NPP -> climate feedback? You need to provide a connection between it and the main research question (NPP -> climate).*

*The next couple of sentences reporting NPP changes in north and south hemispheres are another jump, without apparent connection with the NPP->climate feedback. The last sentence of the first paragraph of Abstract entirely reverses the question: now you are talking about climate -> NPP.*

*It is uncommon that an abstract has more than one paragraph. More importantly, you need a sentence or two to connect the content of the second paragraph to the main issue of this study (NPP -> climate). I think the main point you want to make is that vegetation has a feedback to climate through ET.*

*My main suggestion is that the authors want to first clearly sort out the main logic of this paper. If the main research question is the feedback of NPP to climate, and the main point is that ET is the means of this feedback, then this logic needs to be very clearly presented and all the materials need to be organized around it. Without this big picture in mind, it is easy to get lost in details, e.g., flip back and forth between NPP ->climate and climate -> NPP.*

**Response**: Thanks for the reviewer's good comment. According to the reviewer's comment, we rewrote the abstract as follows:

**Abstract**: The dramatic increase of global temperature since year 2000 has a considerable impact on the global water cycle and vegetation dynamics. Little has

been done about recent feedback of vegetation to climate in different parts of the world, and land evapotranspiration (ET) is the means of this feedback. Here we used the global 1-km MODIS net primary production (NPP) and ET datasets (2000-2014) to investigate their temporospatial changes under the context of global warming. The results showed that global NPP slightly increased in 2000-2014 at a rate of 0.06 $PgC/yr^2$. More than 64% of vegetated land in the Northern Hemisphere (NH) showed increased NPP (at a rate of 0.13 $PgC/yr^2$), while 60.3% of vegetated land in the Southern Hemisphere (SH) showed a decreasing trend (at a rate of -0.18 $PgC/yr^2$). Vegetation greening and climate change promote rises of global ET. Specially, the increased rate of land ET in the NH (0.61 $mm/yr^2$) is faster than that in the SH (0.41 $mm/yr^2$). Over the same period, global warming and vegetation greening accelerate evaporation in soil moisture, thus reducing the amount of soil water storage. Continuation of these trends will likely exacerbate regional drought-induced disturbances and point to an increased risk of drought, especially during regional dry climate phases.

**The main logic of this paper**:

The dramatic increase of global temperature since year 2000 has a considerable impact on the global water cycle and vegetation dynamics. Clear data on spatiotemporal variations and attributes in global terrestrial net primary production (NPP) in different parts of the world within the context of high variability warming are still lacking. Vegetation and climate changes alter the global land evapotranspiration (ET). In our study, we investigated the following three major points of interest: 1) whether the high volatility temperature of the past decade continued to increase NPP, or if different climate constraints were at play; 2) why NPP variations in the Northern and Southern Hemispheres respond differently to climate changes; and 3) what the spatiotemporal variation of NPP is, and what its effects are on ET.

*2) I think the statements you made in the second paragraph are confusing. It seems to me that what you are saying is that no matter NPP increases (in NH) or decreases (in SH), it will lead to drought.*

**Response**: Vegetation greening and climate change promote rises of global ET (Zhang et al., 2015). Specially, the increased rate of land ET in the NH (0.61 $mm/yr^2$)

is faster than that in the SH ($0.41$ mm/yr$^2$). Anomalous warming indicates a general prospective acceleration or intensification of the global hydrological cycle and thus an alteration in the process of ET, but dry conditions have caused a reduction in vegetation productivity and a near cessation of ET growth in the SH.

Figure 5 illustrates the world-wide decrease in soil moisture of four layers (0-10, 10-40, 40-100, and 100-200 cm). Global warming and vegetation greening accelerate evaporation in soil moisture, thus reducing the amount of soil water storage. Continuation of these trends will likely exacerbate regional drought-induced disturbances and point to an increased risk of drought, especially during regional dry climate phases.

References

Zhang, K., Kimball, J.S., Nemani, R.R., Running, S.W., Hong, Y., Gourley, J.J. and Yu, Z.B.: Vegetation greening and climate change promote multidecadal rises of global land evapotranspiration, Sci. Rep, 5, 15956, doi: 10.1038/srep15956, 2015.